# EFFICIENT IMAGE REPRESENTATION LEARNING WITH FEDERATED SAMPLED SOFTMAX

## ABSTRACT

Learning image representations on decentralized data can bring many benefits in cases where data cannot be aggregated across data silos. Softmax cross entropy loss is highly effective and commonly used for learning image representations. Using a large number of classes has proven to be particularly beneficial for the descriptive power of such representations in centralized learning. However, doing so on decentralized data with Federated Learning is not straightforward as the demand on FL clients' computation and communication increases proportionally to the number of classes. In this work we introduce *federated sampled softmax* (*FedSS*), a resource-efficient approach for learning image representation with Federated Learning. Specifically, the FL clients sample a set of classes and optimize only the corresponding model parameters with respect to a sampled softmax objective that approximates the global full softmax objective. We examine the loss formulation and empirically show that our method significantly reduces the number of parameters transferred to and optimized by the client devices, while performing on par with the standard full softmax method. This work creates a possibility for efficiently learning image representations on decentralized data with a large number of classes under the federated setting.

## 1 INTRODUCTION

The success of many computer vision applications, such as classification (Kolesnikov et al., 2020; Yao et al., 2019; Huang et al., 2016), detection (Lin et al., 2014; Zhao et al., 2019; Ouyang et al., 2016), and retrieval (Sohn, 2016; Song et al., 2016; Musgrave et al., 2020), relies heavily on the quality of the learned image representation. Many methods have been proposed to learn better image representation from centrally stored datasets. For example, the contrastive (Chopra et al., 2005) and the triplet losses (Weinberger & Saul, 2009; Qian et al., 2019) enforce local constraints among individual instances while taking a long time to train on $O(N^2)$ pairs and $O(N^3)$ triplets for $N$ labeled training examples in a minibatch, respectively. A more efficient loss function for training image representations is the softmax cross entropy loss which involves only $O(N)$ inputs. Today's top performing computer vision models (Kolesnikov et al., 2020; Mahajan et al., 2018; Sun et al., 2017) are trained on centrally stored large-scale datasets using the classification loss. In particular, using an extremely large number of classes has proven to be beneficial for learning universal feature representations (Sun et al., 2017).

However, a few challenges arise when learning such image representations with the classification loss under the *cross-device* federated learning scenario (Kairouz et al., 2019) where the clients are edge devices with limited computational resources, such as smartphones. First, a typical client holds data from only a small subset of the classes due to the nature of non-IID data distribution among clients (Hsieh et al., 2020; Hsu et al., 2019). Second, as the size of the label space increase, the communication cost and computation operations required to train the model will grow proportionally. Particularly for ConvNets the total number of parameters in the model will be dominated by those in its classification layer (Krizhevsky, 2014). Given these constraints, for an FL algorithm to be practical it needs to be resilient to the growth of the problem scale.

In this paper, we propose a method called *federated sampled softmax* (*FedSS*) for using the classification loss efficiently in the federated setting. Inspired by sampled softmax (Bengio & Senécal, 2008), which uses only a subset of the classes for training, we devise a client-driven negative class

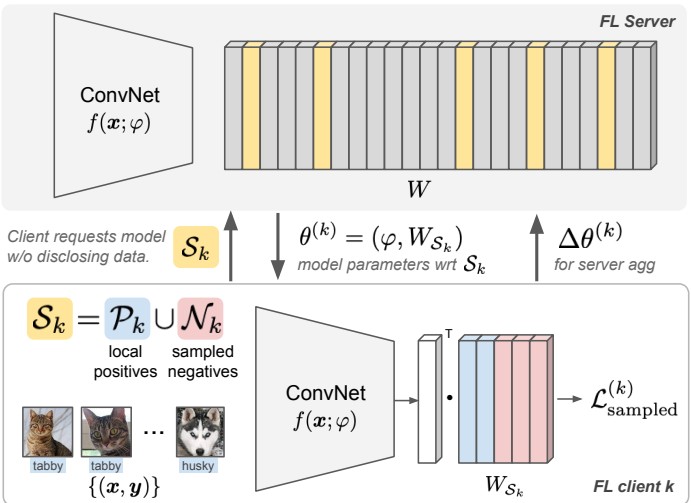

Figure 1: An *FedSS* training round: The client sends a set of obfuscated class labels $\mathcal{S}_k$ to the FL server and receives the feature extractor $\varphi$ and a few columns $W_{\mathcal{S}_k}$, corresponding to classes in $\mathcal{S}_k$, from the weight matrix of the classification layer. The client optimizes this sub network with the sampled softmax loss and then communicates back the model update to the server. The server aggregates the model updates from all the selected clients to construct a new global model for the next round.

sampling mechanism and formulate a sampled softmax loss for federated learning. Figure 1 illustrates the core idea. The FL clients sample negative classes and request a sub network from the FL server by sending a set of class labels that anonymizes the clients' positive class labels in its local dataset. The clients then optimize a sampled softmax loss that involves both the clients' sampled negative classes as well as its local positive classes to approximate the global full softmax objective.

To the best of our knowledge, this is the first work addressing the intersection of representation learning with Federated Learning and resource efficient sampled softmax training. Our contributions are:

1. We propose a novel federated sampled softmax algorithm, which extends the image representation learning via large-scale classification loss to the federated learning scenario.
2. Our method performs on-par with full softmax training, while requiring only a fraction of its cost. We evaluate our method empirically and show that less than $10\%$ of the parameters from the classification layer can be sufficient to get comparable performance.
3. Our method is resilient to the growth of the label space and makes it feasible for applying Federated Learning to train image representation and classification models with large label spaces.

## 2 RELATED WORK

**Large scale classification.** The scale of a classification problem could be defined by the total number of classes involved, number of training samples available or both. Large vocabulary text classification is well studied in the natural language processing domain (Bengio & Senécal, 2008; Liu et al., 2017; Jean et al., 2015; Zhang et al., 2018). On the contrary, image classification is well studied with small to medium number of classes (LeCun et al., 1998; Krizhevsky et al.; Russakovsky et al., 2015) while only a handful of works (Kolesnikov et al., 2020; Hinton et al., 2015; Mahajan et al., 2018; Sun et al., 2017) address training with large number of classes. Training image classification with a significant number of classes requires a large amount of computational resources. For example, Sun et al. (2017) splits the last fully connected layer into sub layers, distributes them on multiple parameter servers and uses asynchronous SGD for distributed training on 50 GPUs. In this work, we focus on a cross-device FL scenario and adopt sampled softmax to make the problem affordable for the edge devices.

**Representation learning.** Majority of works in learning image representation are based on classification loss (Kolesnikov et al., 2020; Hinton et al., 2015; Mahajan et al., 2018) and metric learning objectives (Oh Song et al., 2016; Qian et al., 2019). Using full softmax loss with a large number of classes in the FL setting can be very expensive and sometimes infeasible for two main reasons: (i) exorbitant cost of communication and storage on the clients can be imposed by the classification layer's weight matrix; (ii) edge devices like smartphones typically do not have computational resources required to train on such scale. On the other hand, for metric learning methods (Oh Song et al., 2016; Qian et al., 2019) to be effective, extensive hard sample mining from quadratic/cubic combinations of the samples (Sheng et al., 2020; Schroff et al., 2015; Qian et al., 2019) is typically needed. This requires considerable computational resources as well. Our federated sampled softmax method addresses these issues by efficiently approximating the full softmax objective.

**Federated learning for large scale classification.** The closest related work to ours is Yu et al. (2020), which considers the classification problem with large number of classes in the FL setting. They make two assumptions: (a) every client holds data for a single fixed class label (*e.g.* user identity); (b) along with the feature extractor only the class representation corresponding to the client's class label is transmitted to and optimized by the clients. We relax these assumptions in our work since we focus on learning generic image representation rather than individually sensitive users' embedding. We assume that the clients hold data from multiple classes and the full label space is known to all the clients as well as the FL server. In addition, instead of training individual class representations we formulate a sampled softmax objective to approximate the global full softmax cross-entropy objective.

## 3 METHOD

### 3.1 BACKGROUND AND MOTIVATION

**Softmax cross-entropy and the parameter dominance.** Consider a multi-class classification problem with $n$ classes where for a given input $\boldsymbol{x}$ only one class is correct $\boldsymbol{y} \in [0,1]^n$ with $\sum_{i=1}^{n} y_i = 1$. We learn a classifier that computes a $d$-dimensional feature representation $f(\boldsymbol{x}) \in \mathbb{R}^d$ and logit score $o_i = \boldsymbol{w}_i^T f(\boldsymbol{x}) + b \in \mathbb{R}$ for every class $i \in [n]$. A softmax distribution is formed by the class probabilities computed from the logit scores using the softmax function

$$p_i = \frac{\exp(o_i)}{\sum_{j=1}^{n} \exp(o_j)}, \quad i \in [n]. \tag{1}$$

Let $t \in [n]$ be the target class label for the input $\boldsymbol{x}$ such that $y_t = 1$, the softmax cross-entropy loss for the training example $(\boldsymbol{x}, \boldsymbol{y})$ is defined as

$$\mathcal{L}(\boldsymbol{x}, \boldsymbol{y}) = -\sum_{i=1}^{n} y_i \log p_i = -o_t + \log \sum_{j=1}^{n} \exp(o_j). \tag{2}$$

The second term involves computing the logit score for all the $n$ classes. As the number of classes $n$ increase so does the number of columns in the weight matrix $W \equiv [\boldsymbol{w}_1, \boldsymbol{w}_2, \ldots, \boldsymbol{w}_n] \in \mathbb{R}^{d \times n}$ of the classification layer. The complexity of computing this full softmax loss also grows linearly.

Moreover, for a typical ConvNet classifier for $n$ classes, the classification layer *dominates* the total number of parameters in the model as $n$ increases, because the convolutional layers typically have small filters and the total number of parameters (See Figure 8 in A.1 for concrete examples). This motivates us to use an alternative loss function to overcome the growing compute and communication complexity in the cross-device federated learning scenario.

**Sampled softmax.** Sampled softmax (Bengio & Senécal, 2008) was originally proposed for training probabilistic language models on datasets with large vocabularies. It reduces the computation and memory requirement by approximating the class probabilities using a subset $\mathcal{N}$ of negative classes whose size is $m \equiv |\mathcal{N}| \ll n$. These negative classes are sampled from a proposal distribution $Q$, with $q_i$ being the sampling probability of the class $i$. Using the adjusted logits $o'_j = o_j - \log(mq_j), \forall j \in \mathcal{N}$, the target class probability can be approximated with

$$p'_t = \frac{\exp(o'_t)}{\exp(o'_t) + \sum_{j \in \mathcal{N}} \exp(o'_j)}. \tag{3}$$

This leads to the sampled softmax cross-entropy loss

$$\mathcal{L}_{\text{sampled}}(\boldsymbol{x}, \boldsymbol{y}) = -o'_t + \log \sum_{j \in \mathcal{N} \cup \{t\}} \exp(o'_j). \qquad (4)$$

Note that the sampled softmax gradient is a biased estimator of the full softmax gradient. The bias decreases as $m$ increases. The estimator is unbiased only when the negatives are sampled from the full softmax distribution (Blanc & Rendle, 2018) or $m \to \infty$ (Bengio & Senécal, 2008).

## 3.2 FEDERATED SAMPLED SOFTMAX (FEDSS)

Now we discuss our proposed federated sampled softmax (*FedSS*) algorithm listed in Algorithm 1, which adopts sampled softmax in the federated setting by incorporating negative sampling under FedAvg (McMahan et al., 2017) framework, the standard algorithm framework in federated learning.

One of the main characteristics of FedAvg is that all the clients receive and optimize the exact same model. To allow efficient communication and local computing, our federated sampled softmax algorithm transmits a much smaller sub network to the FL clients for local optimization. Specifcally, we view ConvNet classifiers parameterized by $\theta = (\varphi, W)$ as two parts: a feature extractor $f(\boldsymbol{x}; \varphi) : \mathbb{R}^{h \times w \times c} \to \mathbb{R}^d$ parameterized by $\varphi$ that computes a $d$-dimensional feature given an input image, and a linear classifier parameterized by a matrix $W \in \mathbb{R}^{d \times n}$ that outputs logits for class prediction [1]. The FL clients, indexed by $k$, train sub networks parameterized by $(\varphi, W_{\mathcal{S}_k})$ where $W_{\mathcal{S}_k}$ contains a subset of columns in $W$, rather than training the full model. With this design, federated sampled softmax is more communication-efficient than FedAvg since the full model is never transmitted to the clients, and more computation-efficient because the clients never compute gradients of the full model.

In every FL round, every participating client first samples a set of negative classes $\mathcal{N}_k \subset [n]/\mathcal{P}_k$ that does not overlap with the class labels $\mathcal{P}_k = \{t : (\boldsymbol{x}, \boldsymbol{y}) \in \mathcal{D}_k, y_t = 1, t \in [n]\}$ in its local dataset $\mathcal{D}_k$. The client then communicates the union of these two disjoint sets $\mathcal{S}_k = \mathcal{P}_k \cup \mathcal{N}_k$ to the FL server for requesting a model for local optimization. The server subsequently sends back the sub network $(\varphi, W_{\mathcal{S}_k})$ with all the parameters of the feature extractor together with a classification matrix that consists of class vectors corresponding to the labels in $\mathcal{S}_k$.

---

**Algorithm 1:** Federated sampled softmax (FEDSS). The key differences to the FedAvg are lines 5–7 where the clients request and optimize different sub networks locally. $\eta$ and $\alpha$ are the client and server learning rates, respectively.

1   Initialize $\theta_0 = (\varphi, W)$, where $\varphi$ is the parameter of the feature extractor and $W$ is the classification matrix.
2   **for** each round $t = 0, 1, \ldots$ **do**
3      Select $K$ participating clients.
4      **for** each client $k = 1, 2, \ldots, K$ **do in parallel**
5         Client $k$ samples negatives $\mathcal{N}_k$.
6         Client $k$ requests the model wrt $\mathcal{S}_k = \mathcal{P}_k \cup \mathcal{N}_k$.
7         The server sends back model $\theta_t^{(k)} = (\varphi, W_{\mathcal{S}_k})$.
8         Start local optimization with $\theta^{(k)} \leftarrow \theta_t^{(k)}$.
9         **for** each local mini-batch $b$ over $E$ epochs **do**
10            $\theta^{(k)} \leftarrow \theta^{(k)} - \eta \nabla \mathcal{L}_{\text{sampled}}^{(k)}(b; \theta^{(k)})$
11         $\Delta \theta^{(k)} \leftarrow \theta^{(k)} - \theta_0^{(k)}$
12      $\bar{\boldsymbol{g}}_t \leftarrow \sum_{k=1}^{K} \frac{n_k}{n} \Delta \theta^{(k)}$, where $n = \sum_{k=1}^{K} n_k$
13      $\theta_{t+1} \leftarrow \theta_t - \alpha \bar{\boldsymbol{g}}_t$

---

Then every client trains its sub network by minimizing the following sampled softmax loss with its local dataset

$$L_{\text{FedSS}}^{(k)}(\boldsymbol{x}, \boldsymbol{y}) = -o'_t + \log \sum_{j \in \mathcal{S}_k} \exp(o'_j), \qquad (5)$$

---

[1] We omit the bias term in discussion without loss of generality.

after which the same procedure as FedAvg is used for aggregating model updates from all the participating clients.

In our federated sampled softmax algorithm, the set of positive classes $\mathcal{P}_k$ is naturally constituted by all the class labels from the client's local dataset, whereas the negative classes $\mathcal{N}_k$ are sampled by each client individually. Next we discuss negative sampling and the use of positive classes in the following two subsections respectively.

### 3.3 CLIENT-DRIVEN UNIFORM SAMPLING OF NEGATIVE CLASSES

For centralized learning, proposal distributions and sampling algorithms are designed for efficient sampling of negatives or high quality estimations of the full softmax gradients. For example, Jean et al. (2015) partition the training corpus and define non-overlapping subsets of class labels as sampling pools. The algorithm is efficient once implemented, but the proposal distribution imposes sampling bias which is not mitigable even as $m \to \infty$. Alternatively, efficient kernel-based algorithms (Blanc & Rendle, 2018; Rawat et al., 2019) yield unbiased estimators of the full softmax gradients by sampling from the softmax distribution. These algorithms depend on both the current model parameters $(\varphi, W)$ and the current raw input $\boldsymbol{x}$ for computing feature vectors and logit scores. However, this is not feasible in the FL scenario, one the one hand due to lack of resources on FL clients for receiving the full model, on the other hand due to the constraint of keeping raw inputs only on the devices.

In the *FedSS* algorithm, we assume the label space is known and take a client-driven approach, where every participating FL client uniformly samples negative classes $\mathcal{N}_k$ from $[n]/P_k$. Using a uniform distribution over the entire label space is a simple yet effective choice that does not incur sampling bias. The bias on the gradient estimation can be mitigated by increasing $m$ (See A.3 for an empirical analysis). Moreover, $\mathcal{N}_k$ can be viewed as noisy samples from the maximum entropy distribution over $[n]/P_k$ that mask the client's positive class labels. From the server's perspective, it is not able to identify which labels in $\mathcal{S}_k$ belong to the client's dataset. In practice, private information retrieval techniques (Chor et al., 1995) can further be used such that no identity information about the set is revealed to the server. The sampling procedure can be performed on every client locally and independently without requiring peer information or the current latest model from the server.

### 3.4 INCLUSION OF POSITIVES IN LOCAL OPTIMIZATION

When computing the federated sampled softmax loss, including the set of positive class labels $\mathcal{P}_k$ in Eq. 5 is crucial. To see this, Eq. 5 can be equivalently written as follows (shown in A.6)

$$\mathcal{L}_{\text{FedSS}}^{(k)}(\boldsymbol{x}, \boldsymbol{y}) = \log \left[ 1 + \sum_{j \in \mathcal{S}_k/\{t\}} \exp(o_j' - o_t') \right]. \tag{6}$$

Minimizing this loss function pulls the input image representation $f(\boldsymbol{x}; \varphi)$ and target class representation $\boldsymbol{w}_t$ closer, while pushing the representations of the negative classes $W_{\mathcal{S}_k/\{t\}}$ away from $f(\boldsymbol{x}; \varphi)$. Utilizing $\mathcal{P}_k/\{t\}$ as an additional set of negatives to compute this loss encourages the separation of classes in $\mathcal{P}_k$ with respect to each other as well as with respect to the classes in $\mathcal{N}_k$ (Figure 2d).

Alternatively, not using $\mathcal{P}_k/\{t\}$ as additional negatives leads to a negatives-only loss function

$$\mathcal{L}_{\text{NegOnly}}^{(k)}(\boldsymbol{x}, \boldsymbol{y}) = \log \left[ 1 + \sum_{j \in \mathcal{N}_k} \exp(o_j' - o_t') \right], \tag{7}$$

where $t \in \mathcal{P}_k$ only contributes to computing the true logit for individual inputs, while the same $\mathcal{N}_k$ is shared across all inputs (Figure 2b). Minimizing this negatives-only loss, trivial solutions can be found for a client's local optimization. Because it encourages separation of target class representations $W_{\mathcal{P}_k}$ from the negative class representations $W_{\mathcal{N}_k}$, which can be easily achieved by increasing the magnitudes of the former and reducing those of the latter. In addition, the learned representations can collapse, as the local optimization is reduced to a binary classification problem between the on-client classes $\mathcal{P}_k$ and the off-client classes $\mathcal{N}_k$.

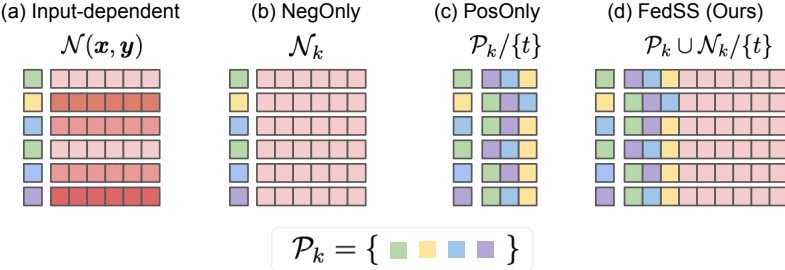

Figure 2: The set of classes providing pushing forces for the local training under different sampled softmax loss formulations. (a) Input-dependent negative classes (depicted by the red squares) are sampled wrt to the inputs and current model, not feasible in the FL setting. (b) Only using the sampled negatives reduces the problem to a binary classification. (c) Using only the local positives lets the local objectives diverge from the global one. (d) FedSS approximates the global objective with sampled negative classes together with local positives.

In contrast, using only the local positives $\mathcal{P}_k$ without the sampled negative classes $\mathcal{N}_k$ gives

$$\mathcal{L}^{(k)}_{\text{PosOnly}}(\boldsymbol{x}, \boldsymbol{y}) = \log \left[ 1 + \sum_{j \in \mathcal{P}_k / \{t\}} \exp(o'_j - o'_t) \right]. \tag{8}$$

Minimizing this loss function solves the client's local classification problem which diverges from the global objective (Figure 2c), especially when $\mathcal{P}_k$ remains fixed over FL rounds and $|\mathcal{P}_k| \ll n$.

## 4 EXPERIMENTS

### 4.1 SETUP

**Notations and Baseline methods.** We denote our proposed algorithm as *FedSS* where both the sampled negatives and the local positives are used in computing the client's sampled softmax loss. We compare our method with the following alternatives:

- *NegOnly*: The client's objective is defined by sampled negative classes only (Eq. 7).
- *PosOnly*: The client's objective is defined by the local positive classes only, no negative classes is sampled (Eq. 8).
- *FedAwS* (Yu et al., 2020): client optimization is same as the *PosOnly*, but a spreadout regularization is applied on server.

In addition, we also provide two reference baselines:

- *FullSoftmax*: The client's objective is the full softmax cross-entropy loss (Eq. 2), serving as performance references when it is affordable for clients to compute the full model.
- *Centralized*: A model is trained with the full softmax cross-entropy loss (Eq. 2) in a centralized fashion using IID data batches.

**Evaluation protocol.** We conduct experiments on two computer vision tasks: multi-class image classification and image retrieval. Performance is evaluated on the test splits of the datasets, which have no sample overlap with the corresponding training splits. We report the mean and standard deviation of the performance metrics from three independent runs. For the *FullSoftmax* and *Centralized* baselines, we report the best result from three independent runs. Please see A.2 for implementation details.

### 4.2 MULTI-CLASS IMAGE CLASSIFICATION

For multi-class classification we use the Landmarks-User-160K (Hsu et al., 2020) and report top-1 accuracy on its test split. Landmarks-User-160k is a landmark recognition dataset created for FL

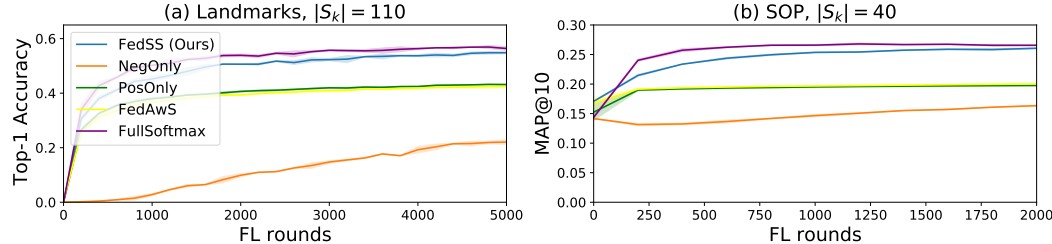

Figure 3: Learning curve for different methods for an average value of number of classes $|\mathcal{S}_k|$ on the clients. The *PosOnly*, *FedAwS* and *FullSoftmax* methods have $|\mathcal{P}_k|$, $|\mathcal{P}_k|$ and $n$ classes respectively, on the clients.

simulations. It consists of 1,262 natural clients based on image authorship. Collectively, every client contains 130 images distributed across 90 class labels. For our experiments $K = 64$ clients are randomly selected to participate in each FL round. We train for a total 5,000 rounds, which is sufficient for reaching convergence.

| $|\mathcal{S}_k|$ | 95 | 100 | 110 | 130 | 170 |
| % of $n$ | (4.7%) | (4.9%) | (5.4%) | (6.4%) | (8.4%) |
|---|---|---|---|---|---|
| FedSS (Ours) | 51.7 ±0.4 | 53.3 ±0.6 | 54.9 ±0.3 | 55.3 ±0.6 | **56.0** ±0.06 |
| NegOnly | 7.1 ±3.7 | 18.7 ±0.4 | 22.0 ±0.8 | 25.0 ±0.4 | 26.5 ±1.4 |
| PosOnly | | | 43.1 ±0.2 | | |
| FedAwS (Yu et al., 2020) | | | 42.5 ±0.4 | | |
| FullSoftmax | | | 56.8 | | |
| Centralized | | | 59.5 | | |

Table 1: Top-1 accuracy (%) on Landmarks-Users-160k at the end of 5k FL rounds. *PosOnly* and *FedAwS* have ∼4.4% of class representations on the clients, whereas, *FullSoftmax* has all the class representations.

Table 1 summarizes the top-1 accuracy on the test split. For *FedSS* and *NegOnly* we report accuracy across different $|\mathcal{S}_k|$. Overall, we observe that our method performs similar to the FullSoftmax baseline while requiring only a fraction of the classes on the clients. Our *FedSS* formulation also outperforms the alternative *NegOnly*, *PosOnly* and *FedAwS* formulations by a large margin. Approximating the full softmax loss with *FedSS* does not degrade the rate of convergence either as seen in Figure 3a. Additionally, Figure 4a shows learning curves for *FedSS* with different $|\mathcal{S}_k|$. Learning with a sufficiently large $|\mathcal{S}_k|$ follows closely the performance of the *FullSoftmax* baseline. We also report performance on ImageNet-21k (Deng et al., 2009) in A.4.

## 4.3 IMAGE RETRIEVAL

| $|\mathcal{S}_k|$ | 25 | 30 | 40 | 60 | 100 |
| % of $n$ | (0.22%) | (0.27%) | (0.35%) | (0.53%) | (0.88%) |
|---|---|---|---|---|---|
| FedSS (Ours) | 25.2 ±0.2 | 25.8 ±0.2 | 26.1 ±0.1 | 26.4 ±0.12 | **26.5** ±0.03 |
| NegOnly | 15.5 ±0.2 | 16.2 ±0.1 | 16.3 ±0.1 | 16.5 ±0.04 | 16.7 ±0.17 |
| PosOnly | | | 19.7 ±0.09 | | |
| FedAwS (Yu et al., 2020) | | | 20.0 ±0.04 | | |
| FullSoftmax | | | 25.7 | | |
| Centralized | | | 25.4 | | |

Table 2: MAP@10 on the SOP dataset at the end of 2k FL rounds.

The Stanford Online Products dataset (Song et al., 2016) has 120,053 images of 22,634 online products as the classes. The train split includes 59,551 images from 11,318 classes, while the test

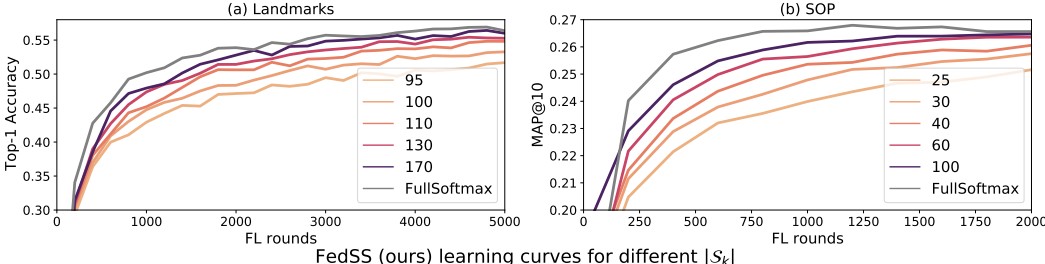

Figure 4: Convergence curves for the proposed *FedSS* method at different cardinalities of $\mathcal{S}_k$. Given that $\mathcal{P}_k$ is fixed for a client, the increase in $|\mathcal{S}_k|$ is caused by increase in $|\mathcal{N}_k|$. The estimate of softmax probability via sampled softmax improves with the increase in $|\mathcal{S}_k|$, and therefore improving the efficacy of the method.

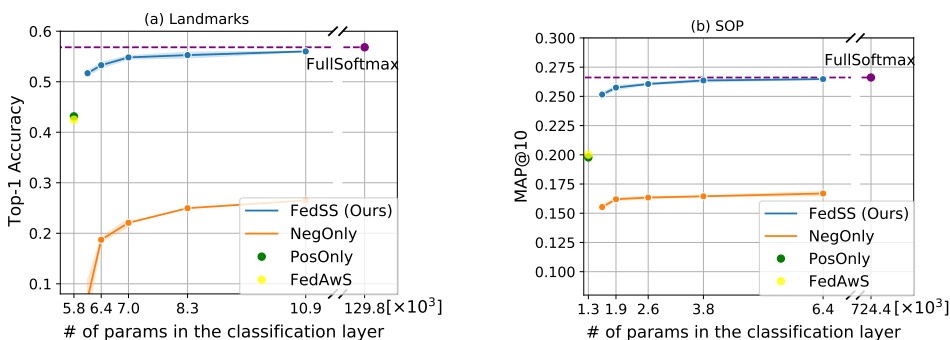

Figure 5: Performance vs number of parameters in the classification layer transmitted to and optimized by the clients for Landmarks-Users-160k (a) and the SOP (b) datasets, respectively.

split includes 11,316 different classes with 60,502 images in total. For FL experiments, we partition the train split into 596 clients, each containing 100 images distributed across 20 class labels. For each FL round, $K = 32$ clients are randomly selected. Similar to metric learning literature, we use nearest neighbor retrieval to evaluate the models. Every image in the test split is used as a query image against the remaining ones. We use normalized euclidean distance to compare two image representations. We report MAP@$R$ ($R = 10$) as the evaluation metric (Musgrave et al., 2020), which is defined as follows:

$$\text{MAP@}R = \frac{1}{R} \sum_{i=1}^{R} P(i), \quad \text{where } P(i) = \begin{cases} \text{precision at } i, & \text{if } i^{\text{th}} \text{ retrieval is correct} \\ 0, & \text{otherwise.} \end{cases} \quad (9)$$

Table 2 summarizes MAP@10 on the SOP test split at the end of 2k FL rounds. Our *FedSS* formulation consistently outperforms the alternative methods while requiring less than $1\%$ of the classes on the clients. This reduces the overall communication cost by 16% when $|\mathcal{S}_k| = 100$ for every client per round. For reasonably small value of $|\mathcal{S}_k|$ our method has a similar rate of convergence to the *FullSoftmax* baseline, as seen in Figure 3b and Figure 4b.

Using the MobilenetV3 (Howard et al., 2019) architecture with embedding size 64, the classification layer contributes to 16% of the total number of parameters in the SOP experiment and 3.4% in the Landarks-User-160k experiment. In the former, our *FedSS* method requires only 84% of the model parameters on every client per round when $|\mathcal{S}_k| = 100$. In the latter, it reduces the model parameters transmitted by 3.38% per client per round when $|\mathcal{S}_k| = 170$ (summarized in Figure 5). These savings will increase as the embedding size or the total number of classes increases (Figure 8 in A.1). For example with embedding size of 1280, which is default embedding size of MobileNetV3, above setup will result in 79% and 38% reduction in the communication cost per client per round for the SOP and Landarks-User-160k datasets, respectively.

## 4.4 ON IMPORTANCE OF $\mathcal{P}_k$ IN LOCAL OPTIMIZATION

One may note that the *NegOnly* loss (Eq. 7) involves fewer terms inside the logarithm than *FedSS* (Eq. 6). To show that the *NegOnly* is not unfairly penalized, we compare the *FedSS* with *NegOnly* such that the number of classes providing pushing forces for every input is the same. This is done by sampling additional $|\mathcal{P}_k| - 1$ negative classes for the *NegOnly* method. As seen in Figure 6, using the on-client classes ($\mathcal{P}_k$) as additional negatives instead of the additional off-client negatives is crucial to the learning.

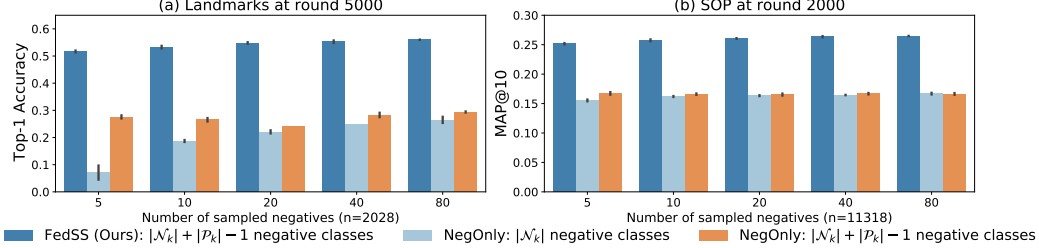

Figure 6: Performance of the *FedSS* (Ours) and *NegOnly* methods with different compositions of the negative classes used for computing the sampled softmax loss. Utilizing on-client classes as additional negatives i.e, *FedSS* method, has superior performance to the *NegOnly* method with equivalent number of negatives.

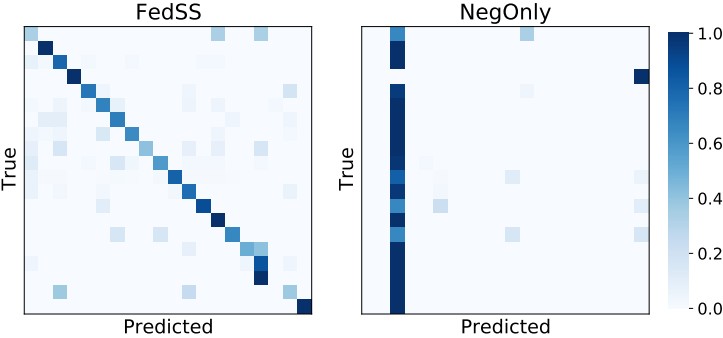

Figure 7: Confusion matrices for $\mathcal{P}_k$ of the same client from Landmarks-User-160k dataset. In both the *FedSS* and *NegOnly* formulations we used $|\mathcal{S}_k| = 95$. In the former, the class representations are learned and well-separated, but are collapsed in the latter.

This boost can be attributed to better approximation of the global objective by the clients. Figure 7 plots a client's confusion matrix corresponding to the *FedSS* and *NegOnly* methods. The *NegOnly* loss leads to a trivial solution for the client's local optimization problem such that the client's positive class representations collapse onto one representation, as reasoned in section 3.4.

## 5 CONCLUSION

Federated Learning is becoming a prominent field of research. Major contributing factors to this trend are: rise in privacy awareness among the general users, surge in amount of data generated by edge devices, and the noteworthy increase in computing capabilities of edge devices. In this work we presented a novel federated sampled softmax method which facilitates efficient training of large models on edge devices with Federated Learning. The clients solve small subproblems approximating the global problem by sampling negative classes and optimizing a sampled softmax objective. Our method significantly reduces the number of parameters transferred to and optimized by the clients, while performing on par with the standard full softmax method. We hope that this encouraging result can inform future research on efficient local optimization beyond the classification layer.

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

# A    Supplementary Material

## A.1    Parameters in the last layer

The number of parameters in the classification layer grows linearly with respect to the number of classes and typically dominates the total number of parameters in the model. Figure 8 shows the number of parameters in the classification layer as the percentage of total number of parameters in the MobileNetV3 model. Each curve shows the percentage for different number of target classes for a fixed embedding size.

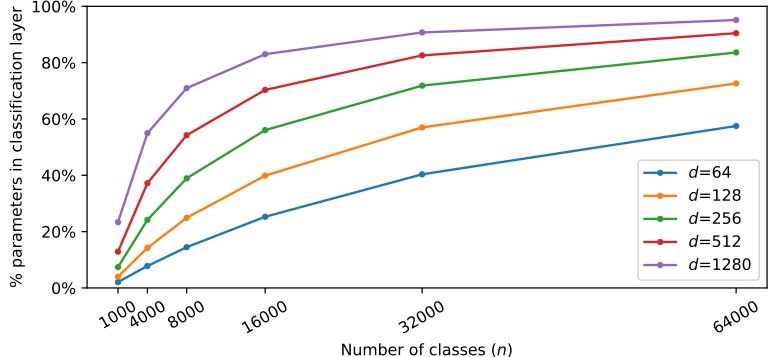

Figure 8: The number of parameters in the classification layer dominates the model as the number of classes $n$ grows. We show the percentage of parameters in the last layer using the MobileNetV3 architecture (Howard et al., 2019) while varying the number of classes $n$ and dimension $d$ of the feature ($d = 1280$ is the default dimensionality of MobileNetV3).

It is obvious that as the number of classes or the size of image representation increases so does the communication and local optimization cost for the full softmax training in the federated setting. In either of these situations our proposed method will facilitate training at significantly lower cost.

## A.2    Implementation Details

For all the datasets we use the default MobileNetV3 architecture (Howard et al., 2019), except that instead of 1280 dimensional embedding we output 64 dimensional embedding. We replace Batch Normalization (Ioffe & Szegedy, 2015) with Group Normalization (Wu & He, 2018) to improve the stability of federated learning (Hsu et al., 2019; Hsieh et al., 2020). Input images are resized to 256×256 from which a random crop of size 224×224 is taken. All ImageNet-21k trainings start from scratch, whereas, for Landmarks-User-160k and the SOP we start from a ImageNet-1k (Russakovsky et al., 2015) pretrained checkpoint. For client side optimization we go through the local data once and use stochastic gradient descent optimizer with batchsize of 32. We use the learning rate of 0.01 for the SOP and Landmarks-User-160k. All ImageNet-21k experiments start from scratch and use the same learning rate of 0.001. To have a fair comparison with *FedAwS* method we do hyperparameter search to find the best spreadout weight and report the performances corresponding to it. For all the experiments, we use scaled cosine similarity with fixed scale value (Wang et al., 2017) of 20 for computing the logits; the server side optimization is done using Momentum optimizer with learning rate of 1.0 and momentum of 0.9. All *Centralized* baselines are trained with stochastic gradient descent.

For a given dataset, all the FL methods are trained for a fixed number of rounds. The corresponding centralized experiment is trained for an equivalent number of model updates.

## A.3    FedSS Gradient noise analysis

Bengio & Senécal (2008) provides theoretical analysis of convergence of the sampled softmax loss. Doing so for the porposed federated sampled softmax within the FedAvg framework is beyond

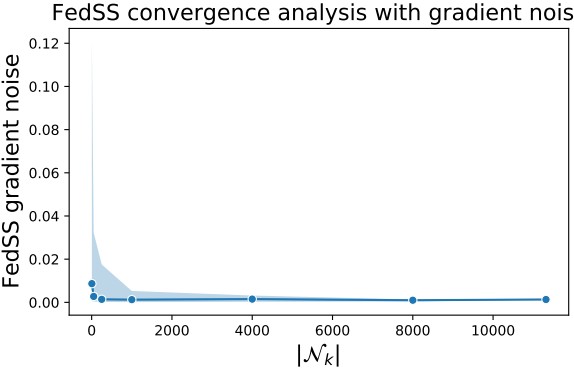

Figure 9: Empirical *FedSS* gradient noise analysis. As we increase the sample size the difference between FedAvg (with *FullSoftmax*) and *FedSS* diminishes.

the scope of this work. Instead we provide an empirical gradient noise analysis for the proposed method. To do so we compute the expected difference between FedAvg (with *FullSoftmax*) and *FedSS* gradients, *i.e.* $\mathbb{E}(|\bar{g}_{FedAvg} - \bar{g}_{FedSS}|)$, where $\bar{g}_{FedAvg}$ and $\bar{g}_{FedSS}$ are client model changes aggregated by the server for FedAvg (with *FullSoftmax*) and *FedSS* methods, respectively. Given that *FedSS* is an estimate of FedAvg (with *FullSoftmax*) this difference essentially represents the noise in *FedSS* gradients.

To compute a single instance of gradient noise we assume that the clients participating in the FL round has same $\mathcal{D}$ with $|\mathcal{D}| = 32$. Please note that the clients will have different $\mathcal{N}_k$. For a given $|\mathcal{N}_k|$ we compute the expectation of the gradient noise across multiple batches ($\mathcal{D}$) of the SOP dataset. Figure 9 shows the *FedSS* gradient noise as a function of $|\mathcal{N}_k|$. For very small values of $|\mathcal{N}_k|$ the gradients can be noisy but as the $|\mathcal{N}_k|$ increases the gradient noise drops exponentially.

### A.4 IMAGENET-21K EXPERIMENTS

Along with Landmarks-User-160K (Hsu et al., 2020) and the SOP (Song et al., 2016) datasets we also experiment with ImageNet-21k (Deng et al., 2009) dataset. It is a super set of the widely used ImageNet-1k (Russakovsky et al., 2015) dataset. It contains 14.2 million images distributed across 21k classes organized by the WordNet hierarchy. For every class we do a random 80-20 split on its samples to generate the train and test splits, respectively. The train split is used to generate 25,691 clients, each containing approximately 400 images distributed across 20 class labels. ImageNet-21k requires a large number of FL rounds given its abundant training images, hence we set a training budget of 25,000 FL rounds to make our experiments manageable. Although the performance we report on ImageNet-21k is not comparable with the (converged) state-of-the-art, we emphasize that the setup is sufficient to evaluate our *FedSS* method and demonstrate its effectiveness.

| $|S_k|$ % of $n$ | 70 (0.3%) | 120 (0.5%) | 220 (1.0%) | 420 (1.9%) | 820 (3.7%) |
|---|---|---|---|---|---|
| FedSS (Ours) | 9.1 ±0.4 | 9.2 ±0.1 | 9.9 ±0.3 | 10.0 ±0.5 | 9.8 ±0.5 |
| NegOnly | 3.9 ±0.1 | 4.2 ±0.1 | 4.3 ±0.2 | 4.4 ±0.1 | 4.7 ±0.2 |
| PosOnly | | | 5.1 ±0.4 | | |
| FedAwS (Yu et al., 2020) | | | 5.1 ±0.1 | | |
| FullSoftmax | | | 11.3 | | |
| Centralized | | | 15.4 | | |

Table 3: Top-1 accuracy (%) on ImageNet-21k at the end of 25k FL rounds. *PosOnly* and *FedAwS* have ∼0.1% of class representations on the clients, whereas, *FullSoftmax* has all the ∼21k class representations.

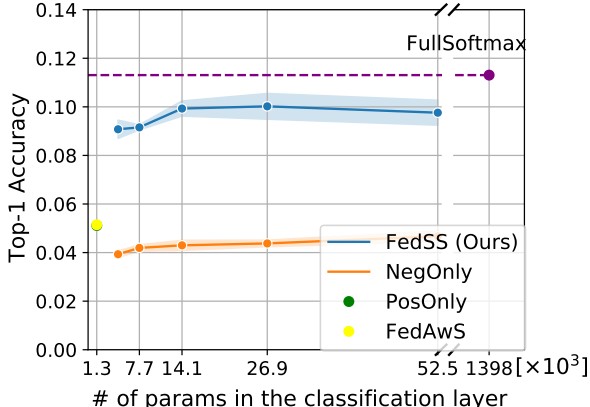

Figure 10: ImageNet-21k: Top-1 accuracy vs number of parameters in the classification layer transmitted to and optimized by the clients.

Table 3 summarizes top-1 accuracy on the ImageNet-21k test split. We experiment with five different choices of $|\mathcal{S}_k|$. The *FullSoftmax* method reaches (best) top-1 accuracy of $11.30\%$ by the end of 25,000 FL rounds, while our method achieves top-1 accuracy of $10.02 \pm 0.5\%$, but with less than 2% of the classes on the clients. Figure 10 summarizess performance of different methods with respect to number of parameters in the classification layer transmitted to and optimized by the clients. Our client-driven negative sampling with positive inclusion method (FedSS) requires a very small fraction of parameters in the classification layer while performing reasonably similar to the full softmax training (FullSoftmax).

## A.5 OVERFITTING IN THE SOP FULLSOFTMAX EXPERIMENTS

The class labels in the train and test splits of the SOP dataset do not overlap. In addition, it has, on average, only 5 images per class label. This makes the SOP dataset susceptible to overfitting (Table 4). In this case, using *FedSS* mitigates the overfitting as only a subset of class representations is updated every FL round.

| Method | Top-1 Accuracy (train) | MAP@10 (test) |
|---|---|---|
| FedSS (Ours) | 97.6 ±0.2 | **26.5** ±0.03 |
| FullSoftmax | 99.9 | 25.7 |
| Centralized | 99.9 | 25.4 |

Table 4: Top-1 accuracy on the train split and corresponding MAP@10 on the test split for the SOP dataset at the end of 2k FL rounds. The *FedSS* shown here is trained on $|\mathcal{S}_k| = 100$.

## A.6 Derivations from Eq. 5 to Eq. 6

*Proof.* Starting from Eq. 5, we have

$$L_{\text{FedSS}}^{(k)}(\boldsymbol{x}, \boldsymbol{y}) = -o_t' + \log \sum_{j \in \mathcal{S}_k} \exp(o_j')$$

$$= \log \left( \exp(-o_t') \cdot \sum_{j \in \mathcal{S}_k} \exp(o_j') \right)$$

$$= \log \sum_{j \in \mathcal{S}_k} \exp(o_j' - o_t')$$

$$= \log \left( \exp(o_t' - o_t') + \sum_{j \in \mathcal{S}_k/\{t\}} \exp(o_j' - o_t') \right)$$

$$= \log \left( 1 + \sum_{j \in \mathcal{S}_k/\{t\}} \exp(o_j' - o_t') \right).$$

This gives Eq. 6. $\qquad\square$

