# OpenReview forum: "Efficient Image Representation Learning with Federated Sampled Softmax"
_ICLR.cc/2022/Conference — ICLR 2022 Submitted_

### Official Review · Reviewer_24vu · 2021-11-01

**Correctness:** 4
**Technical Novelty And Significance:** 2
**Empirical Novelty And Significance:** 2
**Recommendation:** 3
**Confidence:** 4

**Main Review:**

===== Strength ======
1. The paper points out an interesting fact and problem: most of the parameters in a neural net are in the last fully connected (FC) layer, while in a large-number-of-class and non-IID setting, each client may only have data from a small portion of classes. The authors thus accordingly proposed the federated sampled softmax to save the communication cost by local training with only a small number of "negative" classes in addition to the "positive" classes.

2. The authors conduct experiments on three large-scale datasets. The authors conduct the ablation study on how many negative classes are needed.

==== Weakness =====
1. The technical contribution is not sufficient. There have been many methods like (Bengio & Sen´ecal,2008) and [a-c] that subsample the negative classes in the softmax objective. Specifically, in [c], the proposed method did include the positive classes in a minibatch, together with a set of subsampled classes, to compute the softmax. While the authors compare different combinations of positive and negative classes in 3.4, to me, it seems pretty obvious that we should combine the clients' positive classes with subsampled negative classes. Besides, while the authors discussed several advanced methods in 3.3, they ended up choosing uniform sampling. I was wondering if any of these advanced approaches can be approximately implemented in a federated setting to further improve the performance.

[a] Joulin et al., Learning Visual Features from Large Weakly Supervised Data, ECCV 2016

[b] Mikolov et al., Distributed representations of words and phrases and their compositionality, NeurIPS 2013

[c] Hu et al., Learning answer embeddings for visual question answering, CVPR 2018

2. One potential direction to improve the novelty or technical contributions of the paper is to develop a series of methods for different kinds of representation learning in a federated setting, including learning a fully connected layer and metric learning. If we do not consider the computational cost at the client end, will metric learning (without learning the FC layer) outperform the proposed method? In [a], the authors proposed to learn multiple one-vs-all classifiers rather than a softmax classifier. Will it be effective in a federated setting?

3. The experimental setup can be improved. There is no clear description of how many classes, images, local epochs for each dataset. I also have concerns about using the pre-trained features for 4.2 and 4.3, which makes 4.2 and 4.3 like downstream tasks rather than representation learning. I would suggest that the authors include a fixed feature baseline in Table 1 and Table 2 to demonstrate that fine-tuning the feature extractor in a federated setting could outperform the pre-trained features.

4. The authors only apply the proposed method to the FedAvg baseline. Will the proposed method be applicable/effective to more advanced federated learning algorithms like FedProx [d], Scaffold [e], and FedDyn [f]?

[d] Li et al., Federated optimization in heterogeneous networks. In MLSys, 2020

[e] Karimireddy et al., Scaffold: Stochastic controlled averaging for federated learning. In ICML, 2020

[f] Acar et al., Federated learning based on dynamic regularization. In ICLR, 2021

==== Other comments ====
1. I would suggest that the authors add "supervised" into their title, to contrast to many recent works on "unsupervised" representation learning.

2. I would suggest that the authors cite some more papers on federated learning.

**Summary Of The Paper:**

This paper works on "supervised" representation learning in a federated learning setting. The main goal is to save the communication cost: by preventing sending the entire fully-connected layer between the server and the clients if the clients only have data from parts of the classes. The authors proposed federated sampled softmax, which is to compute the softmax only over the "positive" classes of which a client has data and a small portion of the other negative classes. The authors empirically show that, by doing so, even with a very small set of the negative classes (so small communication cost), the resulting feature network can achieve comparable performance to learning with the conventional softmax.

**Summary Of The Review:**

Overall, I enjoy reading the paper as it pointed out an interesting fact and problem. However, the proposed method seems to be too straightforward with insufficient novelty --- negative class subsampling has been widely used and including positive classes of the client seems to be quite intuitive. I have also listed several potential ways to strengthen the paper (please see the main review). I think the current version of the paper is below the acceptance bar, and thus I give a score of "3".

---

> ### Author Response · Authors · 2021-11-20
> **Technical contribution, experimental setup, metric learning and title**
>
> Thank you for the detailed comments. Our response to them is below.
>
> # Technical Contribution
> ## Inclusion of client’s positive labels
> Inclusion of the positive labels seems very intuitive when doing batch based sampled softmax in a centralized setting. FL setting differs in a significant way i.e all the batches have the same N_k, which is not true in the centralized setting as every batch has different sampled negatives. This fixed nature of N_k can have an adverse effect on the FL training. We provide a detailed explanation about this adverse effect in section 3.4 and a solution to resolve it. The proposed method is an outcome of comprehensive experimentation and analysis.
> ## Advanced sampling
> The advanced sampling approaches discussed need access to the full classification matrix to sample from the “true softmax distribution”. This is not possible in our setup as the whole idea is based on not communicating the full softmax matrix. We could send the sample’s embedding and label to the server but this has both privacy and communication consequences. One way of doing advanced sampling is to only send P_k to the server and the server does some kind of hard mining based on the class representation of classes in P_k, though in this case we are not providing any privacy with respect to the client’s labels.
>
> # Experimental setup
> ## Datasets
> All dataset details are provided with exact specifications in sections 4.2, 4.3, A.2 & A.4. We have made a note of providing the number of local epochs which for all our experiments is 1.
> ## Baselines
> We thank the reviewer for this suggestion. For table 2 the fixed feature baseline is 0.1416, we will update table 2 with it. We have made a note of computing and provide it for table 1.
>
> # Metric Learning based representation learning
> Thank you. This is a great idea and worth exploring. Our work is premised on availability of labels and hence softmax classifier, which is a very successful and common method for representation learning, seems a clear choice. We provide additional reasons for choosing softmax  over metric learning methods in section 1 paragraph 1. Exploration of a series of metric learning methods for representation learning is orthogonal to our proposal and beyond the scope of this paper.
>
> # Adding "supervised" into the title
> Thank you for the suggestion. Since more recent literature uses “representation learning” referring to self-supervised representation pre-training, we will consider revising our title to differentiate from those; e.g.“Federated Sampled Softmax: Cross-Device Learning with Large Output Spaces”.

---

> > ### Comment · Reviewer_24vu · 2021-11-29
> > **Thank you for the rebuttal**
> >
> > I thank the authors for the rebuttal. The authors addressed some of my concerns. However, my main concerns on the novelty/contributions are still not fully addressed. No matter in a centralized or a federated setting, adding the positive classes seems quite obvious and straightforward. No further discussion is provided for federated learning algorithms besides FedAvg. Thus, I will keep my score unchanged.
> > Also, I would suggest the authors make a table of the dataset to make the setup more clear.

---

### Official Review · Reviewer_q79o · 2021-11-01

**Correctness:** 4
**Technical Novelty And Significance:** 3
**Empirical Novelty And Significance:** 4
**Recommendation:** 8
**Confidence:** 4

**Main Review:**


# Significance and impact
- The problem studied by the authors has not received a lot of attention, save for the previous work cited by the authors, which was restricted to a narrower problem (1 class per client).
- This paper brings a good practical solution for this problem, building on previous work, which makes it quite significant.

# Writing and clarity

- The paper is very well written and easy to follow, with helpful figures.
- It could be helpful to provide in appendix a short derivation of the validity of the sampled softmax approximation.

# Quality

The experiments are well conducted, with adapted baselines, and support well the claims of the paper. I have a few questions and suggestions:
- in table 2, the proposed method reaches a better performance than fullSoftmax. How do the authors explain this result?
- The paper mainly focuses on the choice of the classes to include in the sampled softmax, and bring a very good experimental support for the method. It would also have been interesting to study:
  - different NN architectures (only mobilenet v3 is used)
  - what is the effect of the proposed method when the number of local epochs increases? This number remains fixed to 1, leading to a very large number of rounds (2k or 5k);
  - What is the effect of label class heterogeneity on the method? e.g. for the SOP dataset one could study splits with varying heterogeneity.

**Summary Of The Paper:**

This paper investigates the problem of training a good computer vision model in the cross-device federated setting, focusing on classification. For deep networks, when the number of classes gets large ($\geq 10^3$), the number of parameters is dominated by the classification layers, hence scaling proportionally to the number of classes in the network. The authors propose to alleviate the resulting computation and communication burden by using sub-networks for each client, with a shared feature extractor but with only a smaller number of classes, re-using the sampled softmax proposed by Bengio & Senécal 2008. The resulting algorithm, Federated Sampled Softmax (fedSS), is benchmarked on image classification and image retrieval tasks along with baselines and variants. Experimental results demonstrate the validity of the approach.

**Summary Of The Review:**

This paper introduces a novel and simple approach to a practical cross-device problem. The experiments are well conducted and support the claims of the paper. I think this paper should be accepted.

---

> ### Author Response · Authors · 2021-11-20
> **SOP overfitting, model architecture and FL hyperparameters**
>
> Thank you for the encouraging comments. We provide answers to your questions below.
>
> # SOP: Why FedSS is better than FullSoftmax?
> In SOP the training classes and testing classes have no overlap which makes it susceptible to overfitting on the training classes. FedSS naturally has an element of regularization as not all class representations are used for computing the gradients. We also provided evidence and a detailed explanation for this behavior in section A.5.
>
> # Model Architecture
> The primary focus of our method is loss formulation to reduce the communication cost of the last linear classifier layer which is proportional to the number of classes. This makes our method independent of the choice of backbone architecture. And using a fixed Mobilenet-V3 architecture seems reasonable given the fact that we are focusing on cross-device FL scenario.
>
> # FL hyperparameters
> ## Local epochs
> We used a simplified setting and avoided introducing hyper parameters like number of local epochs and reporting goals to outline the effectiveness of the proposed method. The behavior of these hyper parameters will be in sync with what has been reported in the literature. In particular, using local epochs might increase the convergence rate wrt FL rounds but might degrade the performance (McMahan et al., 2017).
> ## Label Heterogeneity
> Similarly, we used an extreme non-iid setup to outline the effectiveness of the proposed method. Increase in label heterogeneity will lead to increase in |P_k| and hence better convergence as explained in sections 3.4 and 4.4.

---

### Official Review · Reviewer_nvFP · 2021-11-01

**Correctness:** 3
**Technical Novelty And Significance:** 2
**Empirical Novelty And Significance:** 2
**Recommendation:** 3
**Confidence:** 4

**Main Review:**

Pros:
1. The paper is generally well written and easy to follow.
2. FedSS provides a potential solution to reduce the communication cost of the classification layer in federated learning settings.

Cons:
1. Firstly, the proposed method only has a significant saving when the number of classes is huge (Figure 8, >>1000). However, the datasets used in this paper mostly have a limited number of classes (SOP and Landmarks). In this case, the saving is very small (smaller than 15%). On real tasks with many classes (ImageNet-21k), the proposed method failed to match the accuracy compared to the full Softmax (Table 3). And the increase of $|S_k|$ does not seem to close the gap. Therefore, it is questionable if the proposed algorithm has real-life benefits.
2. The proposed method seems to be a direct application of the sampled Softmax algorithm under a federated learning setting, which restricts the novelty.
3. The proposed method degrades the accuracy (even in Table 1, the results do not really match). How it behaves compared to just using a smaller model with the overall same model size? Since the communication savings on SOP and Landmarks are small, the smaller model baseline should also have roughly the same performance.

**Summary Of The Paper:**

In this paper, the authors proposed federated sampled softmax (FedSS) for resource-efficient federated image representation learning. When the number of classes is large, the final classification layer could take up a large part of the communication cost during training. FedSS allows subsampling the weights of negative classes to reduce data transfer and leads to similar accuracy compared to the full softmax.

**Summary Of The Review:**

The paper is generally well written. However, the experimental results and technical novelty are less convincing. I would lean towards objection for now.

---

> ### Author Response · Authors · 2021-11-20
> **Real life benefits, novelty and accuracy vs communication tradeoff**
>
> Thank you for the detailed comments. Our response to them is below.
> # Real life benefits
> The moderate savings (per round) reported in medium scale datasets might sound trivial but over the whole Federated Learning cycle it will lead to significant savings. We agree that for Imagenet-21k the setting where only 3.7% of classes are communicated to the clients does not match the full softmax setting, but based on our experiments and analysis and Sampled Softmax literature we are confident that as we increase this % it will get close to the full softmax performance. There is no one ideal % for all datasets. The ideal % is the function of the dataset in use. For example, for SOP 0.5-1% is more than enough, whereas, for landmarks it is ~10%. Not having an answer to this upfront does not reduce the real life benefits of the method. In a way this method could be used to efficiently find the ideal %.
>
> # Novelty
> We agree that sampled softmax is not new, but the fact that we provide a detailed algorithm and sampling scheme for adopting sampled softmax in the federated setting is novel in FL literature and opens the door for further research in this direction including server based hard label mining and optimization.
>
> # Accuracy, communication and model architecture
> Our method focuses on reducing the communication cost incurred by the last linear classification layer due to the large label space. Note that FedSS only changes the linear classifier and loss function and is independent of the backbone architecture. Therefore keeping the backbone fixed while varying the sample size offers a fair protocol to evaluate the method. It is also worth noting that using smaller backbone models does not change the fact that the linear classifier layer has a dxN matrix. In fact, the parameters in the last layer would become more dominant in the overall model, where our method would only yield a bigger gain in communication reduction rate.

---

> > ### Comment · Reviewer_nvFP · 2021-11-25
> > **Response**
> >
> > 1. The author argues that the accuracy of ImageNet-21k will be close to the full softmax performance when increasing the % of classes. However, there is no real theoretical or experimental proof to support that. The current good results are only obtained on smaller-scale datasets with fewer classes. In machine learning, the phenomenon could be quite different at different scales. Thus it is unknown if we can really scale up the method to a dataset with many classes (which the paper is targeting for). Since there is no theoretical proof, experimental evidence would be necessary.
> > Even we can recover the accuracy on ImageNet-21k by increasing %, what would be a sweet point of the %? It is possible that we need a very large % (e.g., 80%) to retain the accuracy (if ever possible). In such cases, the saving would be much smaller compared to existing methods like gradient compression (e.g., TernGrad, Deep Gradient Compression).
> >
> > 2. The authors argue that we can simply use the single and same backbone (MobileNetV3) for comparison. I agree with Reviewer UuaD that extra backbones are needed to show that the generalization ability of the algorithm.
> > The experiments also lack the comparison to several important baselines:
> > a. Comparing with gradient compression methods (e.g., TernGrad, Deep Gradient Compression). The author mentioned that "TernGrad quantizes gradients to improve efficiency but all the weights are still required to be transferred when adapting it for FL" in reply to Reviewer UuaD. Nonetheless, TernGrad can reduce the gradient to 2bits, which is 16x smaller compared to full-precision weights. It is not clear if the proposed method can outperform TernGrad at the same amount of transferred data. TernGrad also has a better convergence proof compared to the proposed method. I would still think a comparison is necessary to show the advantage.
> > b. Comparing with using a smaller model. Using a smaller model can also reduce the transferred data by reducing $d$. Reducing the model size and class % can both reduce transfer, but it is unclear which one has a better accuracy-cost trade-off, especially when the authors do not provide results of different backbones (I doubt that using a smaller model will easily provide a good trade-off on SOP and landmark dataset). It is also unclear whether the two dimensions can be well combined.

---

### Official Review · Reviewer_UuaD · 2021-11-03

**Correctness:** 3
**Technical Novelty And Significance:** 2
**Empirical Novelty And Significance:** 2
**Recommendation:** 3
**Confidence:** 3

**Main Review:**

Pros,

This paper is well-written and easy to follow.

The idea is simple.

The proposed method can reduce the communication cost

Cons,

The most significant weak point is that the proposed algorithm is not free from the privacy issue. Since every sampled set of each client has to include the classes that the client has, the central server can infer the classes the client has. In many federated learning applications, it is strictly prohibited that the server can reveal any information of clients.

It would be much better to provide how much the algorithm can save the communication cost. Since the main contribution of this work is not improved performance but reduced communication cost, the authors should provide more discussions about the communication cost.

The authors should discuss with other communication efficient algorithms since the main contribution of this work is saving the communication cost. For instance, there are many works for efficient distributed learning, e.g., TernGrad.

Experiments should include more results considering other data sets and neural net architectures. Since this is not a theory paper, the proposed algorithm can be justified only by experiments. Therefore, ICLR papers are expected to have comprehensive experimental results.


**Summary Of The Paper:**

This paper considers the federated learning problem, especially for the case that there are many classes and each client has a small set of classes. When there are many classes, the neural network has to have many parameters to define the last layer classifier, which makes a huge communication cost. To resolve this issue, the author proposes FedSS, by which each FL client samples a set of classes and defines the loss function only with the sampled classes.  Since unsampled classes are not involved in the client's training process, the client sends only the parameters corresponding to the sampled classes. The authors empirically show the proposed approach can reduce the communication cost without sacrificing performance.


**Summary Of The Review:**

This paper should be very careful to define a class subsampling for federated learning applications since it can leak clients' private information. Also, this paper requires more comprehensive experiments with considering other communication-efficient algorithms.

---

> ### Author Response · Authors · 2021-11-20
> **Privacy and experimental setup**
>
> Thank you for the detailed comments. Our response to them is below.
>
> # Privacy Issue
> We recognize that the method is not full proof. But in a practical setup the user’s positive labels will be changing over time and the two consecutive selections of the client in FL round will have different S_k sets and will not have discernable patterns to expose the client's positive labels. To further increase the noise in S_k a client could randomly drop some of the positive labels. A Bernoulli distribution with predefined *p* value could be used for deciding whether to add a sampled negative label or a positive label to S_k. This has privacy vs training efficiency tradeoff as, as *p* increases so will the noise and hence the training efficiency will reduce.
> In addition, as mentioned in section 2 paragraph 3 we are proposing this method for generic image embedding rather than user sensitive embeddings as in  Yu et al. (2020). Please note that FedAwS does not provide any label privacy.
>
> # Reporting communication cost
> The last paragraph of section 4.3 provides details about reductions in the communication cost. In addition, figures 5 & 10 also outline the communication savings. In all tables we listed the number of classes S_k which directly translates to the communication size. We will also include a table summarizing the communication savings in the Appendix.
>
> # Comparison with other methods
> In FL literature the closest method to ours is the FedAwS for which we provide detailed comparison. Both ours and FedAwS could be seen as gradient sparsification approaches to reduce the communication. TernGrad quantizes gradients to improve efficiency but all the weights are still required to be transferred when adapting it for FL.
>
> # Datasets and architectures
> We carefully selected the datasets to cover two bases i) Number of classes ii) Number of samples. Landmarks-Users-160k covers a moderate number of classes and samples. SOP covers a large number of classes but a small number of samples and Imagenet-21k (reported in Appendix) covers a large number of classes and samples. With respect to model architectures, given that our method focuses only on the loss and the last linear layer of the model, therefore is independent from the choice of backbone neural net architecture. And to provide realistic communication cost and accuracy in the cross-device FL scenario we chose Mobilenet-v3 architecture.

---

### Decision · Program_Chairs · 2022-01-20

**Decision:**

Reject

**Comment:**

The paper revisits representation learning for extreme settings (large number of class categories) in a federated learning setup. The authors show how each client can sample a set of negative classes and optimize only the corresponding model parameters with respect to a sampled softmax objective that approximates the global full softmax objective. The authors investigate the interest of the approach for image classification and image retrieval.

The reviewers appreciated the interest of the approach to reduce communication and the experimental evaluation on several datasets. The reviewers also expressed concerns about privacy, a central concern in federated learning. One reviewer noted for instance that ‘since every sampled set of each client has to include the classes that the client has, the central server can infer the classes the client has’. The reviewers would also have liked to see a more comprehensive evaluation, in the absence of the theoretical guarantees. Finally, the reviewers expressed regarding accuracy/efficiency trade-offs, one reviewer commenting that “the proposed method degrades the accuracy”.

The authors submitted responses to the reviewers' comments. The authors discussed the challenges related to privacy. The authors also commented on other gradient sparsification communication-reducing competing approaches (FedAwS) and the choice of datasets. After reading the response, updating the reviews, and discussion, the reviewers found that ‘the current good results are only obtained on smaller-scale datasets with fewer classes [while] in machine learning, the phenomenon could be quite different at different scales’ and that ‘it is not clear if the proposed method can outperform TernGrad at the same amount of transferred data [and] TernGrad also has a better convergence proof compared to the proposed method’.

We encourage the paper to pursue their approach further taking into account the reviewers' comments, encouragements, and suggestions. Recent progress in privacy protection theoretical frameworks in FL (secure multi party computation, etc.), see the recent survey by Kairouz et al. in FnT in ML, should help the authors develop guarantees for their approach. Moreover the reviewers suggested a clear path towards further improvements of the experimental evaluation.

The revision of the paper will generate a stronger submission to a future venue.

Reject.